# Prevalence, Risk Factors, and Characterization of Multidrug Resistant and ESBL/AmpC Producing *Escherichia coli* in Healthy Horses in Quebec, Canada, in 2015–2016

**DOI:** 10.3390/ani10030523

**Published:** 2020-03-20

**Authors:** Maud de Lagarde, John M. Fairbrother, Julie Arsenault

**Affiliations:** 1OIE Reference Laboratory for *Escherichia coli*, Faculty of Veterinary Medicine, Université de Montréal, Saint-Hyacinthe, QC J2S2M2, Canada; maud.de.lagarde@umontreal.ca; 2Epidemiology of Zoonoses and Public Health Research Unit (GREZOSP), Faculty of Veterinary Medicine, Université de Montréal, Saint-Hyacinthe, QC J2S2M2, Canada; julie.arsenault@umontreal.ca

**Keywords:** antimicrobial resistance, beta-lactamase, cephalosporinase, microbiota, North America, equine

## Abstract

**Simple Summary:**

Antimicrobial resistance has been recognised as a global threat by the WHO. ESBL/AmpC genes, responsible for cephalosporin resistance, are particularly worrisome. *Escherichia coli* is a ubiquitous bacterium. Most strains are commensal, although some can cause disease in humans and animals. Due to its genome plasticity, it is a perfect candidate to acquire resistance genes. We hypothesized that multidrug-resistant *E. coli* and *E. coli* resistant to cephalosporins are present in the fecal microbiota of healthy horses in Quebec. We characterised antimicrobial resistance, identified ESBL/AmpC genes and assessed potential risk factors for their presence. Fecal samples from 225 horses, distributed in 32 premises, were cultured for indicator *E. coli* (selected without enrichment) and specific *E. coli* (selected after enrichment with ceftriaxone). Of the 209 healthy horses in which *E. coli* were detected, 46.3% shed multidrug-resistant (resistant to three or more classes of antimicrobials tested) *E. coli*. Non-susceptibility was most frequently observed for ampicillin, amoxicillin/clavulanic acid or streptomycin. ESBL/AmpC genes were detected in *E. coli* from 7.3% of horses and 18.8% of premises. The number of staff and equestrian event participation within the last three months were identified as risk factors for horses shedding multidrug-resistant *E. coli* isolates. The horse intestinal microbiota is a reservoir for ESBL/AmpC genes. The presence of ESBL/AmpC in horses is both a public and equine health concern, considering the close contact between horses and owners.

**Abstract:**

Although antimicrobial resistance is an increasing threat in equine medicine, molecular and epidemiological data remain limited in North America. We assessed the prevalence of, and risk factors for, shedding multidrug-resistant (MDR) and extended-spectrum β-lactamase (ESBL) and/or AmpC β-lactamase-producing *E. coli* in healthy horses in Quebec, Canada. We collected fecal samples in 225 healthy adult horses from 32 premises. A questionnaire on facility management and horse medical history was completed for each horse. Indicator (without enrichment) and specific (following enrichment with ceftriaxone) *E. coli* were isolated and tested for antimicrobial susceptibility. The presence of ESBL/AmpC genes was determined by PCR. The prevalence of isolates that were non-susceptible to antimicrobials and to antimicrobial classes were estimated at the horse and the premises level. Multivariable logistic regression was used to assess potential risk factors for MDR and ESBL/AmpC isolates. The shedding of MDR *E. coli* was detected in 46.3% of horses. Non-susceptibility was most commonly observed to ampicillin, amoxicillin/clavulanic acid or streptomycin. ESBL/AmpC producing isolates were detected in 7.3% of horses. The most commonly identified ESBL/AmpC gene was *bla_CTX-M-1_*, although we also identified *bla_CMY-2_*. The number of staff and equestrian event participation were identified as risk factors for shedding MDR isolates. The prevalence of healthy horses harboring MDR or ESBL/AmpC genes isolates in their intestinal microbiota is noteworthy. We identified risk factors which could help to develop guidelines to preclude their spread.

## 1. Introduction

Antimicrobial resistance was reported by the World Health Organization (WHO) in 2014 as the largest current threat for global health [1]. Equine medicine is also involved, indeed, the first bacteria resistant to antimicrobials in horses were reported in 1971, in Canada [2]. Subsequently, the number of treatment failure reports due to antimicrobial resistance has increased [3,4,5]. In Europe, several studies have reported that healthy horses can carry multidrug resistant (MDR) bacteria at a relatively high prevalence (39% to 44%) [6,7] and some countries are setting up surveillance monitoring [8]. Nonetheless, molecular and epidemiological data in this species are still limited in North America. In the global approach to antimicrobial resistance recommended by the WHO, horses have been classified as companion animals, although they are also working animals and livestock and could contaminate their owner through direct contact, or even the general population via the food chain. Thus, horses have been overlooked in the general approach to antimicrobial resistance [9].

*Escherichia coli* is ubiquitous and mainly commensal in the intestinal microbiota of mammals. However, pathogenic strains have been recognized, mostly in human and in food-producing animals, and occasionally in horses [10]. Due to its ubiquity, recurrent exposure to systemic (oral, intramuscular or intravenous) antimicrobial treatment and the fast evolution of its genome, this bacterium is considered by the Canadian Integrated Program for Integrated Surveillance System (CIPARS) as an excellent indicator for antimicrobial resistance surveillance [11].

One of the main mechanisms of resistance in *E. coli* is the production of extended spectrum β-lactamases (ESBL) and/or AmpC cephalosporinases (AmpC) [12], resulting in the hydrolyzation of the β-lactam ring, which is present in penicillins, cephalosporins and monobactams. β-lactamase genes (*bla*) have spread very effectively among numerous species of Gram-negative bacteria over the last 30 years [12], both in animals and in humans. In horses, phenotypic resistance to ceftiofur, a third-generation cephalosporin, has been reported in many clinical situations [13]. *bla_CTX-M-1_* is the ESBL resistance gene variant most often detected [14]. However, other variants of CTX-M (i.e., *bla_CTX-M-2_*, *bla_CTX-M-9_*, *bla_CTX-M-14_*, *bla_CTX-M-15_*) have been recognized. *bla_CMY_* and *bla_SHV-12_* have also been identified [14]. All of these variants have also been found in other animal species [14] and in humans [15]. These genes spread mainly through plasmids, carrying multiple resistance genes. Thus, these plasmids convey resistance to other antimicrobial classes, promoting multidrug resistance [16]. However, the resistance gene dissemination can also be enhanced through “high-risk” clones [17]. An example of such a clone is the *E. coli* sequence-type ST410 [18], recently emerging as a public health concern in the human population.

Moreover, owning a horse has been demonstrated as a risk factor for the carriage of ESBL in people [19] in the Netherlands. Even though the author of this study nuanced these results by stating that horse owners often own other pets, and the Netherlands has a high population density which might not be representative of the situation of other countries, nevertheless, this study underlines the potential concern for human health. The colonization with ESBL-producing Enterobacteriaceae, in humans, has been associated with an increase in the length of hospitalization in ICU patients [20]. New regulations restricting the use of antimicrobials such as fluoroquinolones and cephalosporins, classified as having the highest priority by the WHO and Health Canada [21,22], to cases where the veterinarian can prove that there is no better alternative [23], came into effect in early 2019 in veterinary medicine in Canada. Nevertheless, the use of ceftiofur will likely remain common in horses due to the lack of a better alternative, especially for neonatal sepsis and respiratory diseases in adults, possibly enhancing the dissemination of ESBL/AmpC genes.

No data are available on the presence of MDR or ESBL/AmpC-producing isolates in the healthy equine population in Quebec. Our objective was to estimate the prevalence of, and risk factors for, shedding MDR- and/or ESBL/AmpC-producing *E. coli* isolates in horses. We characterized potential ESBL/AmpC isolates for antimicrobial susceptibility and the presence of ESBL/AmpC-associated resistance genes.

## 2. Materials and Methods

### 2.1. Sampling and Data Collection

During the summer 2015, MDL sampled healthy horses from a convenience sample of premises owned by clients and located within a one-hour drive from the CHUV, a university veterinary hospital located in Saint Hyacinthe, Quebec, Canada. To increase the number of sampled horses, in April 2016, the 111 Quebec association of equine veterinary practitioners (AVEQ) members were invited to a conference introducing the project. This event took place in Saint Hyacinthe, Quebec, and was also given in a videoconference. The veterinarians were solicited to sample healthy horses in the stables they visited as part of their veterinary practice. To evaluate the number of targeted horses sampled we used the following equation
*n* = (Z^2^ × P(1 − P))/L^2^
where *n* = the number of targeted horses (*n* = 359 horses), Z = the value from the normal distribution corresponding to the 95% confidence interval (Z = 1.96), and P = the expected prevalence of MDR *E. coli* in the healthy horse population which we extrapolated from a previous article in Great Britain [6] (P = 0.37), and L the desired precision (L = 0.05). Given the large size of the total horse population in Quebec (estimated at 129,000 individuals by Equine Canada in 2010), we have not adjusted the number of horses for a finite population. This figure does not consider the potential non-independence of the status of horses in the same premises.

Every participating veterinarian receive d a sampling kit, including 100 rectal swabs (BBL^TM^ CultureSwab^TM^ Plus, Becton Dickinson, France) and the material to ship the samples to the Ecl laboratory at 4 °C, within 48 h of collection. Protocols were explained in detail in the kit. Each veterinarian could sample up to 10 horses per premise up to a maximum of 10 premises, until the overall target sample size was reached. Only horses over two years old and considered healthy by their owner were eligible for the study. We focused our study on adult horses because breeding does not take a huge place in equestrian activity in Quebec (around 1% of horse riders are interested in breeding in Quebec according to the Cheval Quebec activity report in 2016 (https://cheval.quebec/Rapport-annuel)), therefore we expect that most contacts between people and horses during these activities are with adult horses. The sampled horses were not necessarily part of the veterinarian clientele. Each owner agreed to participate on a voluntary basis. The protocol was approved by the Université de Montréal Ethic Committee for use of animals (15-Rech-1800).

For each sampled horse, the owner and the recruiting veterinarian each completed a questionnaire online, through the Surveymonkey web platform (https://www.surveymonkey.com). They were available in both French and English and are found in the Appendix A of this article (Appendix A). Questions were based on previously reported and suspected risk factors in the horse [24] and were related to the facility management and horse medical history. Each premise was geocoded based on its 6-digit postal code, performed in GeoPinpoint suite version 6.4 (DMTI Spatial).

### 2.2. Indicator Collection: Non-enriched Culture, Antimicrobial Susceptibility Testing, ESBL/AmpC Gene Identification, and Prevalence Estimation

On reception at the Ecl Laboratory, rectal swabs were held in Luria-Bertani (LB) broth for a maximum of 15 min at room temperature, then 100 µL of LB broth was transferred on MacConkey agar and incubated at 37 °C overnight. All lactose-positive colonies, up to a maximum of three, were selected for each rectal sample and cultured in LB broth then plated on MacConkey agar to ensure purity. Isolates were confirmed as *E. coli* by the presence of the *uidA* gene [25], as detected by PCR. Each sample and isolate were stored in 15% glycerol at −80 °C.

Isolates were tested for susceptibility to the 14 antimicrobial agents examined in the CIPARS using the disk-diffusion assay. We used the same disks and techniques as described for the indicator collection of our previous work [7].

When isolates were non-susceptible (intermediate or resistant) to 3rd generation cephalosporins, we looked for 5 β-lactamase resistance genes (*bla_SHV_*, *bla_TEM_*, *bla_CMY-2_*, *bla_OXA_*, *bla_CTX-M_*) by multiplex PCR. We used the same DNA extraction, PCR protocols and CTX-M-variant identification protocols as described for the indicator collection of our previous work [7].

We estimated the prevalence and 95% confidence intervals of (1) horses shedding non-susceptible (i.e., resistant or intermediate) isolate(s) for each antimicrobial, and (2) horses shedding isolate(s) non-susceptible to ≥ 1, 3, 5, 7 and 9 classes of antimicrobials. An isolate was considered MDR if non-susceptible to at least one agent in three or more antimicrobial classes [26]. We used the same method of calculation (with adjustment for sampling weights and clustering within premises) and the same software as described for in the indicator collection of our previous work [7]. We also estimated these prevalences and 95% confidence intervals at the premises level, as previously described [7]; for each outcome, a positive status was attributed when the premises housed at least one positive horse.

### 2.3. Potential ESBL/AmpC Producing E. coli Collection: Culture, Antimicrobial Susceptibility Testing, ESBL/AmpC and Virulence Gene Identification and Descriptive Statistics

ESBL/AmpC-producing bacteria may be shed in small quantities in healthy individuals [27]. To improve detection sensitivity and allow for a more accurate estimation of the proportion of positive horses, we carried out enrichment with ceftriaxone [8,27]. For each rectal swab suspension in LB broth, 1 mL was inoculated in 9 mL of MacConkey broth containing 1 mg/L of ceftriaxone and incubated overnight at 37 °C. When bacterial growth was positive, 100 μL of MacConkey broth was inoculated on MacConkey agar containing 1 mg/L of ceftriaxone and incubated at 37 °C overnight. All isolates up to a maximum of five lactose-positive isolates per sample were selected. All isolates of this collection were confirmed as *E. coli* by the presence of the *uidA* gene [25] as detected by PCR and were tested for susceptibility to 14 antimicrobials, as described above. All isolates in this collection were systematically examined for the presence of five β-lactamase resistance genes (*bla_SHV_*, *bla_TEM_*, *bla_CMY-2_*, *bla_OXA_*, *bla_CTX-M_*) by multiplex PCR (PCR and gene identification protocols are described above).

Descriptive statistics were used to present the non-susceptibility patterns of isolates from this collection. We estimated prevalence with 95% confidence intervals of horses shedding ESBL/AmpC isolates and of the premises housing these horses, using the same calculation method described above.

### 2.4. MDR and ESBL/AmpC: Risk Factors

For the risk factor analyses, two outcome variables were investigated: MDR and ESBL/AmpC status for each horse. A positive MDR or ESBL/AmpC status was defined as the detection of at least one MDR or ESBL/AmpC isolate, respectively, for that horse. All potential risk factors from the questionnaire were categorized. Putative risk factors with *p* < 0.20 (Wald test) in univariable multilevel (facilities, horses) logistic regressions were selected for inclusion in a full multivariable multilevel model for each outcome. Pairwise associations between these selected variables were assessed by χ^2^ test; in the presence of significant association (*p* < 0.05), only one of two correlated variables was kept based on the biological relevance with the outcome. The final multivariable model was refined by sequentially omitting variables with *p* > 0.05 (Wald test). Analyses were performed in MLwiN version 2.36 using 2nd order penalized quasi-likelihood estimation, with no extrabinomial variation permitted. The fit of the final model was evaluated by visual assessment of standardized residuals at the premise level against normal scores and against fixed part prediction.

## 3. Results

In 2015, MDL sampled 67 horses distributed in 10 premises. Following the conference, in April 2016, 14 equine practitioners agreed to participate in the study. Although samples were collected one year apart, the results are presented together as the sampling was similar and there was no modification in the equine practice in Quebec from 2015 to 2016.

A total of 225 horses were sampled, distributed in 32 premises, as illustrated in Figure 1. Between two and 12 horses were sampled in each premise, with a mean of seven horses sampled.

Among the sampled horses, 48% were geldings, 49% were female and 3% were stallions. The mean age was 12 years old with a range from 2 to 30 years old.

### 3.1. Indicator Collection

*E. coli* isolates were detected in 209 (93%) of the 225 rectal swabs. A total of 609 *E. coli* isolates were selected from 209 samples, originating from the 32 premises.

The prevalence estimates of horses shedding non-susceptible isolates per antimicrobial and of the premises housing those horses are shown in Figure 2. Over 40% of horses shed isolates that were non-susceptible to ampicillin, streptomycin or amoxicillin + clavulanic acid. Over 60% of premises housed horses that shed isolates non-susceptible to streptomycin, nalidixic acid, folate pathway inhibitors (trimethoprim-sulfamethoxazole and sulfisoxazole), ampicillin, amoxicillin + clavulanic acid or tetracycline.

As illustrated in Figure 2, in this collection, non-susceptibility to third generation cephalosporins (ceftiofur and ceftriaxone), was observed in 12.8% of horses and 46.8% of premises. We did not identify any *bla* genes as tested by PCR in these isolates.

The prevalence estimate of non-susceptibility to nalidixic acid, a first-generation quinolone, was high (24.7% of horses and 59.4% of premises). In contrast, the estimated prevalence of non-susceptibility to ciprofloxacin, a fluoroquinolone, was relatively low (1.0% of horses and 6.3% of premises).

Prevalence estimates (%) of horses shedding isolates non-susceptible to ≥ 1, 3, 5, 7 or 9 classes of antimicrobials and premises housing these horses, are summarized in Table 1. The prevalence of horses shedding isolates non-susceptible to at least one antimicrobial and MDR isolates were high (80.0% and 46.3%, respectively). Of the 32 premises, 81.3% housed at least one horse shedding MDR isolates. In addition, 1.4% of horses shed isolates non-susceptible to nine classes of antimicrobials, and therefore had a potential for extensive resistance [26].

### 3.2. ESBL/AmpC Collection

A total of 7.3% [95% CI 0–17.6] of the 209 horses shed ESBL/AmpC isolates non-susceptible to ceftriaxone, therefore belonging to the ESBL/AmpC collection, and 18.8% [95% CI 4.5–33] of the 32 premises housed these horses.

Non-susceptibility pattern of the 74 isolates of this collection originating from the 17 positive horses, found in six premises, is shown in Figure 3. All isolates were non-susceptible to ampicillin and ceftriaxone, although three isolates presented susceptibility to ceftiofur and 60 isolates presented susceptibility to cefoxitin, a cephamycin, also considered as a second-generation cephalosporin [28].

Non-susceptibility to aminoglycosides (gentamicin and streptomycin), tetracycline, folate inhibitors (trimethoprim-sulfonamides, sulfizoxasole) and chloramphenicol were present in over 60% of the isolates.

A total of 54.1% of isolates were non-susceptible to a first-generation quinolone (nalidixic acid) and 20.3% of isolates were non-susceptible to ciprofloxacin, a fluoroquinolone, in this collection. These isolates were therefore non-susceptible to two families of antimicrobial classified as having the highest priority in human medicine by both Canadian Health and the WHO [21,22].

The main ESBL genes identified were *bla_CTX-M-1_* (43/74 tested isolates) and *bla_SHV_* (15/74), four isolates carried a combination of *bla_CTX-M-1_* and *bla_SHV_*. Nine isolates carried the AmpC gene *bla_CMY-2_*. In four isolates we could not detect tested ESBL/AmpC genes by PCR.

### 3.3. Risk Factors

A total of 13 potential risk factors were derived from the questionnaire (Table 2). Eleven were considered at the individual level and two were considered at the premise level.

Data with missing values, representing almost half of the dataset, were excluded from modeling.

A total of five risk factors were selected for multivariate modeling (all *p* < 0.20 in univariable logistic regressions) for the MDR outcome. The variable “The horse presented an infection” was then excluded as it was associated with “The horse has been medically treated within the last 3 months”. The variable “Transportation within the last 3 months” was excluded because it was associated with “Participating in an equestrian event within the last 3 months”.

According to the multivariate model, the odds of being an MDR horse were 3.5 times higher (*p* = 0.03) among the horses that had participated to an equestrian event within the last three months and 3.4 times higher (*p* = 0.01) if the horse was in a premise where the staff were composed of more than five persons (Table 3). Visual assessment of residuals at the premise level suggested that our model fitted the data.

Interaction between the two variables of the final multivariable model were checked but were not significant (*p* = 0.41, Wald test) and thus not kept in the model.

For the ESBL/AmpC outcome, considering the high percentage of missing data and low number of positive horses, no statistical modelling was performed.

## 4. Discussion

The present study illustrates that the fecal microbiota of healthy horses in Quebec, Canada, harbor MDR and ESBL/AmpC *E. coli* isolates. The prevalence of horses shedding ESBL/AmpC *E. coli* isolates (7.3%) is comparable to that which was detected phenotypically in the United Kingdom in 2012 (6.3%) [6]. Nevertheless, at the premise level, it seems that the prevalence in Quebec (18.8%) is inferior to the prevalence reported in France (29.0%) [7]. However, these regional differences in apparent prevalence might be related to a higher sensitivity in the detection of positive premises in the study in France, considering that in France we tested more horses per premise (between six and 36 horses per premises) and ESBL/AmpC isolates were detected by two enrichment methods. The prevalence we found in horses in Quebec contrasts with the 1% prevalence observed in Ontario among 188 healthy dogs in 2009 [29], although, even if the calculation methods are not the same, this is still lower than the 26.5% of fecal carriage of ESC-resistant Enterobacteriaceae in healthy dogs in Ontario in 2018 [30]. The prevalence of horses shedding ESBL/AmpC *E. coli* isolates in Quebec is higher than that reported in Sable Island horses, where 1/508 horses shedding an ESBL gene [31] (*bla_CTX-M-1_*) was found. This is not surprising because our horse population is in contact with the populations of other species in which ESBL/AmpC genes have been detected, such as pigs [32], poultry [33], cattle [34] and humans [35], underlining the importance of the one health approach [1,36] to address the problem.

Our study reported the presence of isolates that are non-susceptible to nine classes of antimicrobial in an indicator collection of *E. coli* from horses for the first time to our knowledge, which is worrisome. Although these isolates may be commensals, it is possible that putative resistance genes are carried by mobile genetic elements, such as plasmids, and are therefore transmissible to potential pathogenic or zoonotic strains. The dissemination of extensive resistance to pathogenic strains could lead to an increased risk of complications in the treatment of infections caused by these strains.

Enrofloxacin, a fluoroquinolone, is classified as having a very high importance in human medicine [22] and is approved for veterinary use in equine medicine. Resistance to quinolones is known to be acquired and is mostly due to the apparition of chromosomal mutations, although resistance genes carried by plasmids have also been reported [37]. Often, the chromosomal mutations appear consecutively and are localized on the genes *gyrA* and *parC* (coding for gyrase and topoisomerase, respectively, both involved in the DNA synthesis). The number of mutations is proportionate to the minimal inhibitory concentration (the more the higher). Hence, non-susceptibility to nalidixic acid is generally precursory for fluoroquinolone treatment failure [38]. In the indicator collection, we detected 24.7% of horses and 59.4% of premises presenting a non-susceptibility to nalidixic acid, suggesting that enrofloxacin should be used with caution, to maintain its efficacy in horses in Quebec. In the ESBL/AmpC collection, we found 14 isolates presenting a non-susceptibility to both 3^rd^ generation cephalosporin and fluoroquinolones. Even though these isolates are unlikely to be pathogenic, they still represent a risk of dissemination due to their high capacity to resist antimicrobial pressure. They could acquire virulence genes through the transfer of plasmids thus becoming a threat for public and/or equine health.

In our study, the predominant ESBL gene found was *bla_CTX-M-1_*. ESBL of the CTX-M family have become a public health concern in the last two decades, their incidence and diversity having increased dramatically during this time and have overridden other ESBL variants such as *bla_TEM_* and *bla_SHV_* in gram negative bacteria [15]. The *bla_CTX-M_* encoded ESBL family is characterized by the ability to inhibit 3rd and 4th generation cephalosporins and monobactams, but not cephamycins and carbapenems. These ESBLs are also known to be susceptible to β-lactam inhibitors. However, no cephamycin or penicillin/β-lactam inhibitor combinations are approved or used off-label (to the authors’ knowledge) to treat horses. The predominance of *bla_CTX-M-1_* suggests a global dissemination of this gene in the equine population both in Europe and in North America. The absence of other variants of *bla_CTX-M_* in the Quebec horse population contrasts with the high diversity of *bla_CTX-M_* found in the healthy equine population in France and throughout Europe [7]. This suggests that the presence of this family of genes may have occurred later in North America than in Europe, and that the genes may not yet have had the time to diversify.

We detected the AmpC resistance gene *bla_CMY-2_* in several horses. This gene has been frequently found in poultry and pigs, including in Quebec [32,39]. Although this gene has previously been identified in one healthy horse in France, the fact that we identified it in several healthy horses in Quebec suggests the possibility of AmpC gene spread between animal species. Indeed, horses can be in contact with other animal species, including dogs, cats, poultry among others, in the premises.

Even though we detected 12.8% of horses carrying isolates non-susceptible to 3rd generation cephalosporins in the indicator collection, none of these isolates carried the tested ESBL/AmpC genes, similar to what had been found in the indicator collection of our previous work [7]. These findings suggest that other mechanisms of resistance to cephalosporins (for example, alteration of the protein binding protein) may be present in the population. These alternative mechanisms are less likely to spread through plasmids but could impact cephalosporin efficacy, and therefore could affect equine welfare. We also found four isolates of the ESBL/AmpC collection in which we could not identify a *bla* gene. This could indicate that other, less common, *bla* genes are present in the horse population.

Among the risk factors model selected for modeling, the correlation between the variable “The horse presented an infection” and “The horse has been medically treated within the last 3 months” was to be expected, because a horse with an infection is often treated for this infection. The medical treatment of the horse was considered more biologically relevant to influence the shedding of MDR *E. coli* rather than the infection itself. However, this variable was not retained in the final model, perhaps because of the absence of specific information about the type of treatment, which could include treatments other than antimicrobials.

A correlation between “Transportation within the last 3 months” and “Participating in an equestrian event within the last 3 months” was also observed, which was not surprising, as horses which participate in an equestrian event are often transported to the equestrian event. We chose to consider participation in the equestrian event because of the possibility of transmission of antimicrobial resistance genes inter- and intraspecies during the event.

To our knowledge, we demonstrated for the first time that participation in an equestrian event was a risk factor for shedding MDR isolates at the horse level. Considering the correlation between the horse participation in an equestrian event and transportation, this effect could also be driven by contacts occurring during transportation. Based on this association, we could suggest isolating horses that are participating in equestrian events or at least the implementation of appropriate biosecurity measures. As an example, limiting contact between these horses and horses that stay at home or handling horses that stayed at home before horses that travelled might be beneficial to limit antimicrobial gene dissemination. However, more longitudinal studies are needed to establish the duration of shedding, and therefore be more accurate in these recommendations.

Our results suggest that a higher number of persons taking care of horses daily increases the risk of detecting MDR isolates in the horse’s intestinal microbiota. We previously documented that this factor was associated with a higher risk of detecting ESBL/AmpC isolates in the healthy equine population in France [7]. The fact that this variable was found to be significant in both studies is noteworthy. Indeed, such information is easily obtained, and therefore could be helpful for elaborating guidelines to improve equine health. It could help equine veterinarians in defining “at-risk” equine populations and encourage the use of antimicrobial susceptibility testing in these populations.

The absence of a probability sampling method in our study might affect the representativeness of our prevalence estimate. The extrapolation of such estimate to the general equine population should be made cautiously, as the horses selected for our study are more likely representative of a subpopulation of horses under regular veterinary follow-up examination. Another limitation of our study is the recruitment of a smaller sample size of horses than planned, combined with a high percentage of missing data for the questionnaire among recruited horses (almost 50%), thus reducing the precision of the prevalence estimates and statistical power of the risk factor analyses. The low participation rate could be due to a lack of awareness of the importance of antimicrobial resistance in the equine industry. A higher proportion of missing values were present in horses shedding MDR isolates. This could be due to some regional differences and/or owner characteristics influencing both the risk of MDR and interest to participate in our study. The validity of our results depends on the absence of association between response rate and exposure to identified risk factors. Such association seems unlikely considering that the MDR status and associated risk factors were unknown for both horse owners and veterinarians at the time of data collection.

A valuable follow up to this study would be to sample the veterinarians and owners of these horses and see if there is a correlation between horses and horse handlers for the carriage of ESBL/AmpC producing *E. coli*. Another interesting follow up would be to repeat the study a few years after the regulations (see introduction) have been set up and see if these have made a difference.

## 5. Conclusions

In conclusion, we found a noteworthy prevalence of ESBL/AmpC genes and MDR isolates in the fecal microbiota of healthy horses in Quebec. Surveillance of ESBL/AmpC gene dissemination and the quantification of MDR isolates would be beneficial to characterize the nature and the extent of the risk they represent, with the aim of limiting their transmission between horses, but also to other species including humans and to the environment. The detection of risk factors for MDR shedding could be used to help equine veterinarians in managing at-risk populations.

## Figures and Tables

**Figure 1 animals-10-00523-f001:**
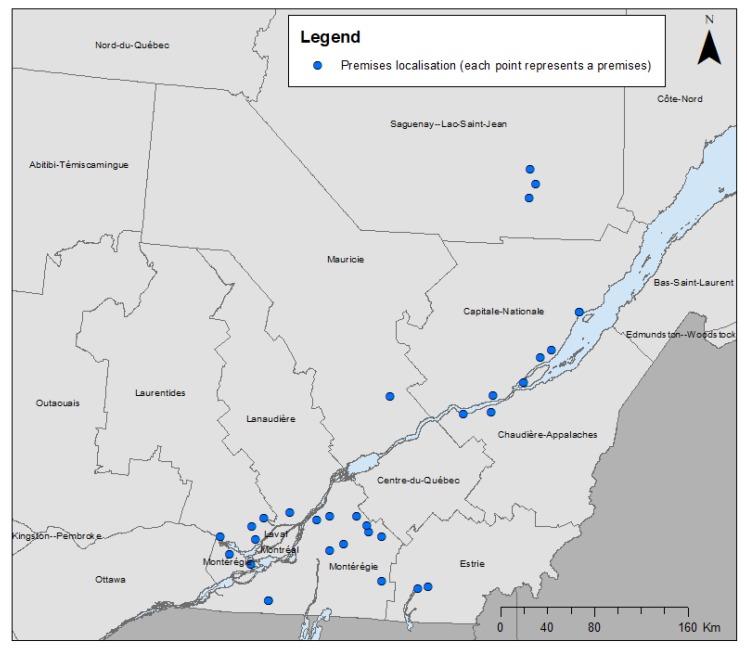
Geographical distribution of the sampled premises (based on center point of their 6-digit postal code) over the administrative regions of the province of Quebec in a cross-sectional study of 209 healthy adult horses in 32 premises performed in 2015 and 2016. Two premises in Capitale-Nationale and two premises in Monteregie were very close, and therefore are overlapping on the map. Mapping was performed in ArcGIS version 10.6, using reference maps from Statistics Canada (2016 census).

**Figure 2 animals-10-00523-f002:**
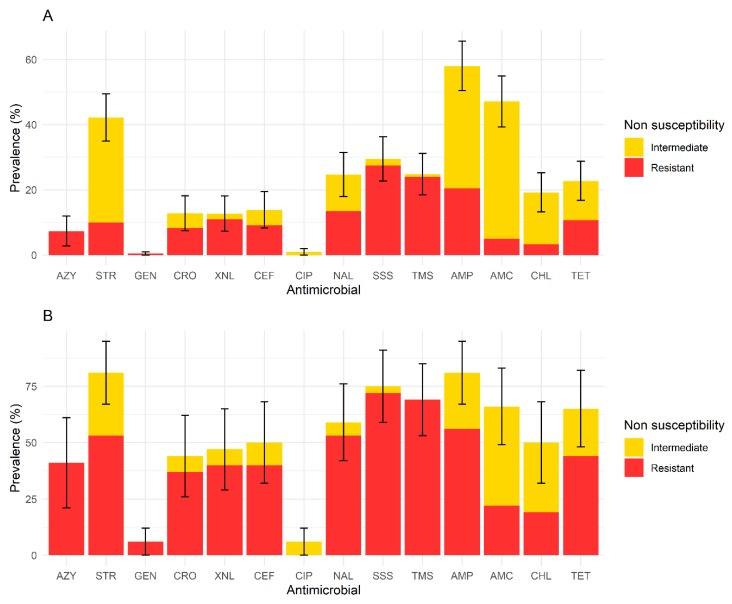
Prevalence estimates (%) of non-susceptibility (yellow and red) for each antimicrobial, at the horse level (**A**), and at the premises level (**B**), in a cross-sectional study of 209 healthy adult horses, in 32 premises, performed in 2015 and 2016, in Quebec. Bars represent 95% confidence intervals for prevalence of non-susceptible isolates. The proportion of resistant isolates for each antimicrobial is presented in red. A total of 609 isolates were tested. Abbreviations: AZY = azithromycin, STR = streptomycin, GEN = gentamicin, CRO = ceftriaxone, XNL = ceftiofur, CEF = cefoxitin, CIP = ciprofloxacin, NAL = nalidixic acid, SSS = sulfisoxazole, TMS = trimethoprim–sulfamethoxazole, AMP = ampicillin, AMC = amoxicillin/clavulanic acid, CHL = chloramphenicol, TET = tetracycline.

**Figure 3 animals-10-00523-f003:**
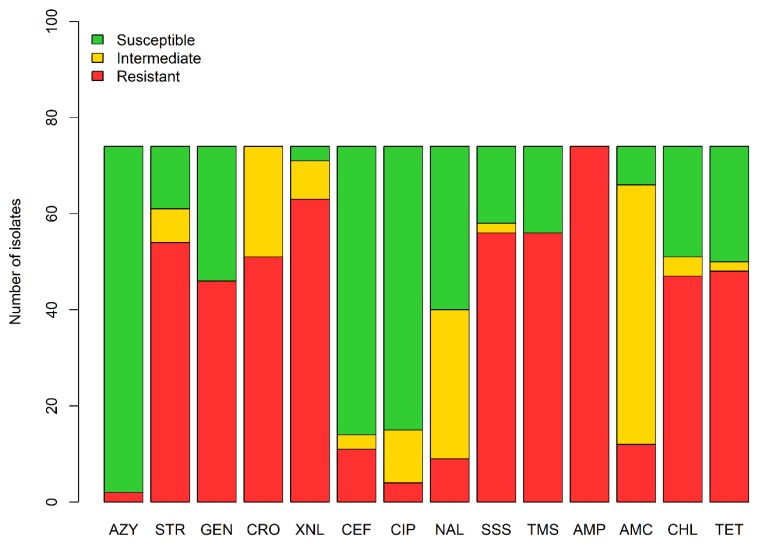
Susceptibility profiles of *E. coli* isolates in the ESBL/AmpC collection, in a cross-sectional study performed on healthy adult horses, in Quebec in 2015 and 2016 (n = 74 isolates distributed in 17 horses among 6 premises). Abbreviations: AZY = azithromycin, STR = streptomycin, GEN = gentamicin, CRO = ceftriaxone, XNL = ceftiofur, CEF = cefoxitin, CIP = ciprofloxacin, NAL = nalidixic acid, SSS = sulfisoxazole, TMS = trimethoprim–sulfamethoxazole, AMP = ampicillin, AMC = amoxicillin/clavulanic acid, CHL = chloramphenicol, TET = tetracycline.

**Table 1 animals-10-00523-t001:** Prevalence estimates (%) with 95% confidence intervals (95% CI) of healthy adult horses shedding *E. coli* isolates that are non-susceptible to more than 1, 3, 5, 7 or 9 classes of antimicrobials and premises housing these horses based on the indicator collection results in a cross-sectional study of 209 horses in 32 premises in Quebec in 2015 and 2016. Abbreviations: CI = confidence interval, MDR = multidrug-resistant.

Number of Resistant Antimicrobial Classes	Indicator Collection			
	Horse Level (*n* = 209)		Premises Level (*n* = 32)	
	%	95% CI	%	95% CI
≥1	80.0	69.8–90.2	96.9	90.5–100
≥3 (MDR)	46.3	34.5–58.0	81.3	67.0–95.5
≥5	15.3	8.5–22.3	53.1	34.8–71.4
≥7	3.9	1.5–6.2	25.0	9.1–40.9
≥9	1.4	0–3.2	9.4	0–20.1

**Table 2 animals-10-00523-t002:** Descriptive statistics and *p*-value (Wald test) from univariable multilevel logistic regression analyses of potential risk factors for MDR status in horses in a cross-sectional study performed on healthy adult horses, in Quebec, in 2015 and 2016. In bold are the factors that were retained for the multivariable analysis.

Putative Risk Factors		Number of Horses	% of MDR-Positive Horses	*p*-Value
*Horse-level*				
	**Transportation out of the horse’s premises within the last 3 months**			**0.11**
	Yes	31	41.9	
	No	69	29.0	
	**Participation to an equestrian event within the last 3 months**			**0.13**
-	Yes	19	57.9	
-	No	92	29.3	
	Housing			0.28
-	Stable	46	39.1	
-	Pasture ^1^	64	29.7	
	Activity			0.73
-	Sport (competition)	39	43.6	
-	Leisure	66	30.3	
-	Reproduction	4	25.0	
	The horse presents a chronic disease			0.47
-	Yes	18	22.2	
-	No	123	39.0	
	**The horse presented an infection (diagnosed by the veterinarian) within the last 3 months**			**0.03**
-	Yes	12	75.0	
-	No	124	34.7	
	The horse presented diarrhea within the last 3 months			0.91
-	Yes	5	20.0	
-	No	61	23.0	
	The horse was hospitalized within the last 3 months			The model did not converge
-	Yes	2	100	
-	No	137	37.2	
	The horse has undergone surgery within the last 3 months			The model did not converge
-	Yes	1	100	
-	No	137	38.0	
	**The horse has been medically treated within the last three months (all treatment considered)**			**0.02**
-	Yes	39	53.9	
-	No	94	30.9	
	The horse presented with colic within the last 3 months			0.81
-	Yes	5	40	
-	No	131	37.4	
*Premise-level*				
	Total number of horses in the premises ^2^			0.26
-	Less than 15	101	45.5	
-	15 and more	109	48.6	
	**Number of staff taking care of horses daily ^3^**			**0.01**
-	Less than 5 persons	68	23.5	
-	5 persons and more	38	47.4	

^1^ Defined as a horse that stays on pasture at night and has a shelter in the pasture. ^2^ Categorization was done *a posteriori*, based on the mean of the number of horses in the premises we sampled. ^3^ This variable was already categorized in the questionnaire.

**Table 3 animals-10-00523-t003:** Parameter estimates and odds ratios from a multivariable regression modeling MDR positive status at the horse level, based on the results of a cross-sectional study performed on 32 premises and 209 healthy adult horses, sampled in Quebec, in summers 2015 and 2016. The estimated variance at the premises level was 0.171 (standard error of 0.316).

Risk Factor for the Outcome	Odds Ratios		
	Estimate	95% CI	*p*
The horse has participated in an equestrian event within the last three months (yes vs. no)	3.5	1.1, 11.1	0.03
Number of staff taking care of horses daily (5 persons and more vs. less than 5 persons)	3.4	1.3, 8.7	0.01

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
