# Peer review of "Prevalence, Risk Factors, and Characterization of Multidrug Resistant and ESBL/AmpC Producing Escherichia coli in Healthy Horses in Quebec, Canada, in 2015–2016"

_animals, 2020, doi:10.3390/ani10030523_

Round 1

Reviewer 1 Report

In my opinion, the topic of the article is of great importance. As the frequency of multi-resistant bacteria increases worldwide, it is very important to cover every aspect of animal and human environment. Especially horses play an important role as they are considered pets more than livestock. Therefore, they are in much closer contact to humans than lifestock and the transmission of multi-resistant bacteria from one to another is way more likely.

However, I believe that the presented research is not good enough for this scientific journal. I recommend that authors look for a magazine with less requirements. I motivate my advice with the following premises:

The research contains old results (2015 - 2016). The topic of drug resistance is an issue that is subject to dynamic changes. We are in 2020, the obtained results and conclusions are already 5 years old and this is disqualifying. Research was too scarce. A larger number of genes should be determined, this will give a better picture of drug resistance. Research, of course, is important, but currently a lot is being created on similar subjects and published in magazines with a smaller IF.

Author Response

We agree that antimicrobial resistance is a dynamic topic. However, we think that monitoring the evolution in time is as important as making a description of the current situation. The publication of our results will allow to document a first baseline for monitoring trends in horses. In fact, these data provide a first portrait of the situation in healthy horses in Quebec, and even in Canada. Moreover, the documentation of risk factors, for which the association will likely remains valid in time, is critical for orienting control actions.

We also agree that we could have looked for other antimicrobial genes, however we had very limited financial support, because as mentioned in the paper, horses are very much overlooked in the literature, as far as antimicrobial resistance is concerned, and we selected the antimicrobial genes that appeared to us the most relevant in a clinical context.

Reviewer 2 Report

In this manuscript, de Lagarde et al. examined the prevalence, risk factors and characteristics of multidrug resistant (MDR) and ESBL/AmpC producing E. coli in the fecal microbiota of 225 healthy horses from 32 premises of Quebec, Canada in 2015-2016. MDR E. coli was detected in 46.3% of the horses, and ESBL/AmpC genes were detected in E. coli from 7.3% of the horses. The number of staff and equestrian event participation within the last three months were identified as risk factors for horses shedding MDR E. coli isolates.

This work is straight forward and easy to follow in general. Potential risk factors identified are of potential values to the development of practical guidelines on the control of spread of MDR E. coli among horses. However, I have a few major concerns as follows:

1. Significant amount of text has been borrowed from a previous work by the same authors (Ref 7). To avoid falling into the pitfall of self-plagiarism, the authors are highly recommended to paraphrase relevant text.

2. With an estimated prevalence of only 7.3% for the ESBL/AmpC genes, it is not convincing to claim that the prevalence is a “substantial” one (L48 and L428).

3. In multivariate modelling, between the associated variables, why was one excluded but not the other (L316)? what is the rationale behind? is it based on a smaller p-value or what?

Other concerns:

- L21: specify that it is the “intestinal” or “fecal” microbiota. Also on L429.

- L37: what is the questionnaire about?

- L74: “Gram” is the name of the scientist who developed Gram staining and thus needs to be capitalised

- L77: do you mean “blaCTX-M-14” instead?

- L89: a redundant description already mentioned on L76

- L140: what is the difference between the two?

- L145: is “the protocol” about PCR?

- L153: remove “respectively”

- L167: remove the first “All isolates”

- L192: it is confusing that the whole Results section is in point form. Please reorganise into paragraphs.

- L193: do you mean “April 2015” instead? The conference should precede the study, right?

- L208: sulfisoxazole too

- L228: remove the grid lines in Figure 2

- L234: cefoxitin is actually abbreviated as “CEF” in the figure

- L246: table legend should always go above the body of the table. Also for Table 2 (L307) and Table 3 (L331)

- L258: “cefoxitin” instead

- L288: specify that it is “nalidixic acid”

- L331: how about the remaining risk factor selected for multivariate modelling? a non-significant result is also worth reporting

- L338: what is the exact value for UK?

- L342: how many horses exactly?

- L347: provide a percentage value as well

- L350: provide a reference for the One health concept

- L358-365: provide reference(s) for each of the statements. Also for L374-379.

- L365: define “MIC”

- L393: vs “12.7%” on L218

Author Response

First, we would like to thank the reviewers for their careful reading of our manuscript and their constructive remarks. You can find below the answer to their comments.

In this manuscript, de Lagarde et al. examined the prevalence, risk factors and characteristics of multidrug resistant (MDR) and ESBL/AmpC producing E. coli in the fecal microbiota of 225 healthy horses from 32 premises of Quebec, Canada in 2015-2016. MDR E. coli was detected in 46.3% of the horses, and ESBL/AmpC genes were detected in E. coli from 7.3% of the horses. The number of staff and equestrian event participation within the last three months were identified as risk factors for horses shedding MDR E. coli isolates.

This work is straight forward and easy to follow in general. Potential risk factors identified are of potential values to the development of practical guidelines on the control of spread of MDR E. coli among horses. However, I have a few major concerns as follows:

  1. Significant amount of text has been borrowed from a previous work by the same authors (Ref 7). To avoid falling into the pitfall of self-plagiarism, the authors are highly recommended to paraphrase relevant text.

Answer : The text has been rephrased as much as possible for the introduction and discussion. For the Material and Method we referred to the first article as much as possible to keep the text understandable for the reader. We do not consider there was a possibility self plagiarism in the results paragraphs.

NB : We submitted our text to two plagiarism checker available and none of them found plagiarism in our text.

  1. With an estimated prevalence of only 7.3% for the ESBL/AmpC genes, it is not convincing to claim that the prevalence is a “substantial” one (L48 and L428).

Answer : We changed “substantial” for “noteworthy” which is a less strong word in both lines.

Lines 47-49 now reads : “. Prevalence of healthy horses harboring MDR or ESBL/AmpC genes isolates in their intestinal microbiota is noteworthy”

Lines 439-440 now reads : “In conclusion, we found a noteworthy prevalence of ESBL/AmpC genes and MDR isolates in the fecal microbiota of healthy horses in Quebec”.

  1. In multivariate modelling, between the associated variables, why was one excluded but not the other (L316)? what is the rationale behind? is it based on a smaller p-value or what?

Answer : As recommended in an epidemiology textbook [1] in the “model building strategies” section   when two variables are correlated, and contain similar information, one of them should be excluded based on either biological plausibility, or missing information, or reliability of measurement. As the two other criteria were similar, we excluded variables based on biological relevance. Clarifications were added in the results section.

Line 314-323 now reads : “The variable “The horse presented an infection” was then excluded as it was associated with “The horse has been medically treated within the last 3 months”. The correlation between these two variables was to be expected, because a horse with an infection is often treated for this infection. The medical treatment of the horse was considered more biologically relevant to influence the shedding of MDR E. coli rather than the infection itself. The variable “Transportation within the last 3 months” was excluded because it was associated with “Participating in an equestrian event within the last 3 months”. It is not surprising that these two variables were correlated, as horses which participate in an equestrian event are often transported to the equestrian event, but we chose to consider the participation in the equestrian event because of the possibility of transmission of antimicrobial resistance genes inter and intra species during the event.”

Other concerns:

- L21: specify that it is the “intestinal” or “fecal” microbiota. Also on L429.

Answer : we added “fecal” to the sentence.

Lines 19-21 now reads : “We hypothesized that multidrug resistant E. coli and E. coli resistant to cephalosporins are present in the fecal microbiota of healthy horses in Quebec.”

Lines 435-436 now reads : “In conclusion, we found a noteworthy prevalence of ESBL/AmpC genes and MDR isolates in the fecal microbiota of healthy horses in Quebec”.

- L37: what is the questionnaire about?

Answer : We added clarifications.

Lines 37-38 now reads : “A questionnaire on facility management and horse medical history was completed for each horse.”

- L74: “Gram” is the name of the scientist who developed Gram staining and thus needs to be capitalised

Answer : Line 75 : The capital has been added.

- L77: do you mean “blaCTX-M-14” instead?

Answer : line 79 : we corrected the misprint.

- L89: a redundant description already mentioned on L76

Answer : the comment on ceftiofur has been removed on line 93.

- L140: what is the difference between the two?

Answer : The veterinary CLSI 2015 uses clinical breakpoints from animal pathogens, the CLSI 2015 uses human pathogens. However, this part has been removed du to self-plagiarism and is now referred to our previous work [2].

- L145: is “the protocol” about PCR?

Answer : This section has been removed due to self plagiarism and now is referred to our previous work [2].

- L153: remove “respectively”

Answer : we removed “respectively”.

Lines 148-150 now reads : “We estimated the prevalence and 95% confidence intervals of 1) horses shedding non-susceptible (i.e. resistant or intermediate) isolate(s) for each antimicrobial, and 2) horses shedding isolate(s) non-susceptible to ≥ 1, 3, 5, 7 and 9 classes of antimicrobials.”

- L167: remove the first “All isolates”

Answer : we believe that the meaning of the sentence would be modified, therefore we kept the original sentence.

- L192: it is confusing that the whole Results section is in point form. Please reorganise into paragraphs.

Answer : The whole result section has been reorganised into paragraphs.

- L193: do you mean “April 2015” instead? The conference should precede the study, right?

Answer : We realized the text could mislead the reader in the description of the sampling. We hope the modifications to the text brought clarifications.

Lines 103-107 now reads : “During the summer 2015 MDL sampled healthy horses from a convenience sample of premises owned by clients and located within a one-hour drive from the CHUV, a university veterinary hospital located in Saint-Hyacinthe, Quebec, Canada. To increase the number of sampled horses, in April 2016, the 111 Quebec association of equine veterinary practitioners (AVEQ) members were invited to a conference introducing the project.”

- L208: sulfisoxazole too

Answer : Line 204 : Sulfisoxazole has been added in brackets.

- L228: remove the grid lines in Figure 2

Answer : We thank the reviewer for this suggestion, however we believe the grid lines make the reading of the figure easier so we kept them.

- L234: cefoxitin is actually abbreviated as “CEF” in the figure

Answer : We changed the legend of the figure, to harmonise the abbreviation of cefoxitine

- L246: table legend should always go above the body of the table. Also for Table 2 (L307) and Table 3 (L331)

Answer : The legend were placed before the tables.

- L258: “cefoxitin” instead

Answer : Line 257 : The e has been removed.

- L288: specify that it is “nalidixic acid”

Answer : Nalidixic acid was added into brackets.

Line 287-288 reads now : “54.1% of isolates were non-susceptible to a first-generation quinolone (nalidixic acid) and 20.3% of isolates were non-susceptible to ciprofloxacin, a fluoroquinolone, in this collection”.

- L331: how about the remaining risk factor selected for multivariate modelling? a non-significant result is also worth reporting

Answer : We reported all p values for the univariate analysis. Among the variables initially selected for inclusion in the full multivariable model, a backward selection procedure was applied. Thus, at each step of the selection procedure, one variable was rejected at a time until all remaining variables were statistically significant. It means that p-values of variables still in the model were re-estimated at each step. It is unusual to report estimates and p-values at each step of the selection process, as only the final model is interpreted. All variables not kept in the final model can be interpreted as statistically not significant (or as excluded from the modeling in the case of highly correlated predictors, as we observed for participation to an event and transportation in our study). We reported the results as recommended by an epidemiology textbook [1].

- L338: what is the exact value for UK?

Answer : Line 344 : the value for UK (6.3%) has been added in brackets.

- L342: how many horses exactly?

Answer : Line 347-348 : The number of horses tested per premises in the study in France was provided.

- L347: provide a percentage value as well

Answer : As surprising as it can be there are very few data available on the prevalence of extended-spectrum cephalosporinase (ESC)-producing Escherichia coli in different animal species in Canada. There are several study providing proportions of positive tested isolates but, to our knowledge, none of these numbers can be extrapolated to the general population. Therefore, we prefer not to provide these percentages that can be misleading.

- L350: provide a reference for the One health concept

Answer : Line 356 : A reference to the webpage on One Health by the WHO in 2017 has been added.

- L358-365: provide reference(s) for each of the statements. Also for L374-379.

Answer : The very good reviews on quinolone resistance mechanisms by Jacoby [3] (Line 367) and on blaCTX-M by Canton [4] (Line 382)have been added in the reference.

- L365: define “MIC”

Answer : MIC has been replaced by minimal inhibitory concentration.

Lines 369-370 now reads : “The number of mutations is proportionate to the minimal inhibitory concentration (the more the higher).”  

- L393: vs “12.7%” on L218

Answer : We corrected the value on line 215 to 12.8%

Reviewer 3 Report

Interesting and well written publication about important topic - AMR horses. The methods are sound, the results are clearly written and the discussion is relevant. 

The replies to the previous review are very valuable. I included some small suggestions/notes in the pdf, but overall I think this is an article worth publishing. 

PS one additional publication that might be considered to be included is from C. M. Isgren, 1N. J. Williams, 2O. D. Fletcher, 1,2D. Timofte, 3T. W. Maddox, 3P. D. Clegg and 1G. L. Pinchbeck on AMR surveillance horses UK clinical submissions - so not healthy horses! 

All the best. 

Author Response

First, we would like to thank the third reviewer for their careful reading of our manuscript and their constructive remarks.

Please find below our answers to their comments :

Comment 1 : Correct that this study finds this, but they also make the remark that it could be due to horse owners often owned multiple species of companion animals. Plus took place in the Netherlands, where the population density is very high, and animals/people living closely together. Line 93

Answer 1 : We added a comment and line 87 to 91 now reads : “Moreover, owning a horse has been demonstrated as a risk factor for the carriage of ESBL in people [18] in the Netherlands. Even though the author of this study nuanced these results by stating that horse owners often own other pets, and the Netherlands has a high population density which might not be representative of the situation of other countries, nevertheless, this study underlines the potential concern for human health.”

Comment 2 : Suggest to delete 'Indeed' and just start the sentence with 'The colonization ...

Answer 2 : Line 91 : As suggested we removed the word “Indeed”.

Comment 3 : e.g. for neonatal sepsis and respiratory diseases

Answer 3 : We added the suggested comment and lines 97-98 now reads : “Nevertheless, the use of ceftiofur will likely remain common in horses due to lack of a better alternative, especially for neonatal sepsis and respiratory diseases in adults, possibly enhancing the dissemination of ESBL/AmpC genes.”

Comment 4 : why only adult horses?

Answer 4 : line 128-132 : to better explain the reason why we focused on adult horses, we added the comment : “We focused our study on adult horses because breeding does not take a huge place in equestrian activity in Quebec (around 1% of horse riders are interested in breeding in Quebec according to the Cheval Quebec activity report in 2016 (https://cheval.quebec/Rapport-annuel)), therefore, we expect that most contacts between people and horses during these activities are with adult horses.”

Comment 5 : ??

We are not sure to understand this comment however we tried to make the sentence more clear : line 255 now reads : “Prevalence estimates (%) with 95% confidence intervals (95% CI) of healthy adult horses shedding E. coli isolates that are non-susceptible to more than 1, 3, 5, 7 or 9 classes of antimicrobials”.

Comment 6 : For Ceftiofur we usually use CTF as abbreviation.

Answer 6 : We feel that the XNL abbreviation (based on the commercial name of ceftiofur) is more appropriate for veterinarians on the field, who will hopefully read this article. Moreover, this abbreviation is commonly used in other articles.

Comment 7 : This sentence belongs more to the discussion not the results

Answer 7 : The explanation for the model construction have been moved to the discussion.

Comment 8 : As follow up study, would be interesting to sample the vets and owners and see if there is any correlation. And of the text

Answer 8 : A comment has been added : line 452-455 : “A valuable follow up to this study would be   to sample veterinarians and owners of these horses and see if there is a correlation between horses and horse handlers for the carriage of  ESBL/AmpC producing E. coli . Another interesting follow up would be to repeat the study a few years after the regulations (see introduction) have been set up and see if this made a difference. “

Comment 9 : one additional publication that might be considered to be included is from C. M. Isgren, 1N. J. Williams, 2O. D. Fletcher, 1,2D. Timofte, 3T. W. Maddox, 3P. D. Clegg and 1G. L. Pinchbeck on AMR surveillance horses UK clinical submissions - so not healthy horses! 

Answer 9 : The reference has been added in the introduction, line 61. (although it is surprising that the authors are not cited in the actual article).